# A Full Non-Monotonic Transition System for Unrestricted Non-Projective Parsing

## Abstract

Restricted non-monotonicity has been shown beneficial for the projective arc-eager dependency parser in previous research, as posterior decisions can repair mistakes made in previous states due to the lack of information. In this paper, we propose a novel, fully non-monotonic transition system based on the non-projective Covington algorithm. As a non-monotonic system requires exploration of erroneous actions during the training process, we develop several non-monotonic variants of the recently defined dynamic oracle for the Covington parser, based on tight approximations of the loss. Experiments on datasets from the CoNLL-X and CoNLL-XI shared tasks show that a non-monotonic dynamic oracle outperforms the monotonic version in the majority of languages.

## 1   Introduction

Greedy transition-based dependency parsers are widely used in different NLP tasks due to their speed and efficiency. They parse a sentence from left to right by greedily choosing the highest-scoring transition to go from the current parser configuration or state to the next. The resulting sequence of transitions incrementally builds a parse for the input sentence. The scoring of the transitions is provided by a statistical model, previously trained to approximate an *oracle*, a function that selects the needed transitions to parse a gold tree.

Unfortunately, the greedy nature that grants these parsers their efficiency also represents their main limitation. McDonald and Nivre (2007) show that greedy transition-based parsers lose accuracy to error propagation: a transition erroneously chosen by the greedy parser can place it

in an incorrect and unknown configuration, causing more mistakes in the rest of the transition sequence. Training with a dynamic oracle (Goldberg and Nivre, 2012) improves robustness in these situations, but in a monotonic transition system, erroneous decisions made in the past are permanent, even when the availability of further information in later states might be useful to correct them.

Honnibal et al. (2013) show that allowing some degree of non-monotonicity, by using a limited set of non-monotonic actions that can repair past mistakes and replace previously-built arcs, can increase the accuracy of a transition-based parser. In particular, they present a modified arc-eager transition system where the Left-Arc and Reduce transitions are non-monotonic: the former is used to repair invalid attachments made in previous states by replacing them with a leftward arc, and the latter allows the parser to link two words with a rightward arc that were previously left unattached due to an erroneous decision. Since the Right-Arc transition is still monotonic and leftward arcs can never be repaired because their dependent is removed from the stack by the arc-eager parser and rendered inaccessible, this approach can only repair certain kinds of mistakes: namely, it can fix erroneous rightward arcs by replacing them with a leftward arc, and connect a limited set of unattached words with rightward arcs. In addition, they argue that non-monotonicity in the training oracle can be harmful for the final accuracy and, therefore, they suggest to apply it only as a fall-back component for a monotonic oracle, which is given priority over the non-monotonic one. Thus, this strategy will follow the path dictated by the monotonic oracle the majority of the time. Honnibal and Johnson (2015) present an extension of this transition system with an Unshift transition allowing it some extra flexibility to correct past errors. However, the restriction that only rightward

arcs can be deleted, and only by replacing them with leftward arcs, is still in place. Furthermore, both versions of the algorithm are limited to projective trees.

In this paper, we propose a non-monotonic transition system based on the non-projective Covington parser, together with a dynamic oracle to train it with erroneous examples that will need to be repaired. Unlike the system developed in (Honnibal et al., 2013; Honnibal and Johnson, 2015), we work with full non-monotonicity. This has a twofold meaning: (1) our approach can repair previous erroneous attachments regardless of their original direction, and it can replace them either with a rightward or leftward arc as both arc transitions are non-monotonic;[1] and (2) we use exclusively a non-monotonic oracle, without the interferences of monotonic decisions. These modifications are feasible because the non-projective Covington transition system is less rigid than the arc-eager algorithm, as words are never deleted from the parser's data structures and can always be revisited, making it a better option to exploit the full potential of non-monotonicity. To our knowledge, the presented system is the first non-monotonic parser that can produce non-projective dependency analyses. Another novel aspect is that our dynamic oracle is *approximate*, i.e., based on efficiently-computable approximations of the loss due to the complexity of calculating its actual value in a non-monotonic and non-projective scenario. However, this is not a problem in practice: experimental results show how our parser and oracle can use non-monotonic actions to repair erroneous attachments, outperforming the monotonic version developed by Gómez-Rodríguez and Fernández-González (2015) in a large majority of the datasets tested.

## 2 Preliminaries

### 2.1 Non-Projective Covington Transition System

The non-projective Covington parser was originally defined by Covington (2001), and then recast by Nivre (2008) under the transition-based parsing framework.

---

[1]The only restriction is that parsing must still proceed in left-to-right order. For this reason, a leftward arc cannot be repaired with a rightward arc, because this would imply going back in the sentence. The other three combinations (replacing leftward with leftward, rightward with leftward or rightward with rightward arcs) are possible.

The transition system that defines this parser is as follows: each parser configuration is of the form $c = \langle \lambda_1, \lambda_2, B, A \rangle$, such that $\lambda_1$ and $\lambda_2$ are lists of partially processed words, $B$ is another list (called the buffer) containing currently unprocessed words, and $A$ is the set of dependencies that have been built so far. Suppose that our input is a string $w_1 \cdots w_n$, whose word occurrences will be identified with their indices $1 \cdots n$ for simplicity. Then, the parser will start at an initial configuration $c_s(w_1 \ldots w_n) = \langle [], [], [1 \ldots n], \emptyset \rangle$, and execute transitions chosen from those in Figure 1 until a terminal configuration of the form $\{\langle \lambda_1, \lambda_2, [], A \rangle \in C\}$ is reached. At that point, the sentence's parse tree is obtained from $A$.[2]

These transitions implement the same logic as the double nested loop traversing word pairs in the original formulation by Covington (2001). When the parser's configuration is $\langle \lambda_1 | i, \lambda_2, j | B, A \rangle$, we say that it is considering the **focus words** $i$ and $j$, located at the end of the first list and at the beginning of the buffer. At that point, the parser must decide whether these two words should be linked with a leftward arc $i \leftarrow j$ (Left-Arc transition), a rightward arc $i \rightarrow j$ (Right-Arc transition), or not linked at all (No-Arc transition). However, the two transitions that create arcs will be disallowed in configurations where this would cause a violation of the **single-head constraint** (a node can have at most one incoming arc) or the **acyclicity constraint** (the dependency graph cannot have cycles). After applying any of these three transitions, $i$ is moved to the second list to make $i-1$ and $j$ the focus words for the next step. As an alternative, we can instead choose to execute a Shift transition which lets the parser read a new input word, placing the focus on $j$ and $j+1$.

The resulting parser can generate any possible dependency tree for the input, including arbitrary non-projective trees. While it runs in quadratic worst-case time, in theory worse than linear-time transition-based parsers (e.g. (Nivre, 2003; Gómez-Rodríguez and Nivre, 2013)), it has been shown to outspeed linear algorithms in practice, thanks to feature extraction optimizations that cannot be implemented in other parsers (Volokh and Neumann, 2012). In fact, one of the fastest dependency parsers ever reported uses this algorithm

---

[2]In general $A$ is a forest, but it can be converted to a tree by linking headless nodes as dependents of an artificial root node at position 0. When we refer to parser outputs as trees, we assume that this transformation is being implicitly made.

| | |
|---|---|
| Shift: | $\langle \lambda_1, \lambda_2, j|B, A \rangle \Rightarrow \langle \lambda_1 \cdot \lambda_2|j, [], B, A \rangle$ |
| No-Arc: | $\langle \lambda_1|i, \lambda_2, B, A \rangle \Rightarrow \langle \lambda_1, i|\lambda_2, B, A \rangle$ |
| Left-Arc: | $\langle \lambda_1|i, \lambda_2, j|B, A \rangle \Rightarrow \langle \lambda_1, i|\lambda_2, j|B, A \cup \{j \rightarrow i\} \rangle$ |
| | only if $\nexists k \mid k \rightarrow i \in A$ (single-head) and $i \rightarrow^* j \notin A$ (acyclicity). |
| Right-Arc: | $\langle \lambda_1|i, \lambda_2, j|B, A \rangle \Rightarrow \langle \lambda_1, i|\lambda_2, j|B, A \cup \{i \rightarrow j\} \rangle$ |
| | only if $\nexists k \mid k \rightarrow j \in A$ (single-head) and $j \rightarrow^* i \notin A$ (acyclicity). |

Figure 1: Transitions of the monotonic Covington non-projective dependency parser. The notation $i \rightarrow^* j \in A$ means that there is a (possibly empty) directed path from $i$ to $j$ in $A$.

(Volokh, 2013).

## 2.2 Monotonic Dynamic Oracle

A dynamic oracle is a function that maps a configuration $c$ and a gold tree $t_G$ to the set of transitions that can be applied in $c$ and lead to some parse tree $t$ minimizing the Hamming loss with respect to $t_G$ (the amount of nodes whose head is different in $t$ and $t_G$). Following Goldberg and Nivre (2013), we say that an arc set $A$ is **reachable** from configuration $c$, and we write $c \rightsquigarrow A$, if there is some (possibly empty) path of transitions from $c$ to some configuration $c' = \langle \lambda_1, \lambda_2, B, A' \rangle$, with $A \subseteq A'$. Then, we can define the loss of configuration $c$ as

$$\ell(c) = \min_{t|c \rightsquigarrow t} \mathcal{L}(t, t_G),$$

and therefore, a correct dynamic oracle will return the set of transitions

$$o_d(c, t_G) = \{\tau \mid \ell(c) - \ell(\tau(c)) = 0\},$$

i.e., the set of transitions that do not increase configuration loss, and thus lead to the best parse (in terms of loss) reachable from $c$. Hence, implementing a dynamic oracle reduces to computing the loss $\ell(c)$ for each configuration $c$.

Goldberg and Nivre (2013) show a straightforward method to calculate loss for parsers that are **arc-decomposable**, i.e., those where every arc set $A$ that can be part of a well-formed parse verifies that if $c \rightsquigarrow (i \rightarrow j)$ for every $i \rightarrow j \in A$ (i.e., each of the individual arcs of $A$ is reachable from a given configuration $c$), then $c \rightsquigarrow A$ (i.e., the set $A$ as a whole is reachable from $c$). If this holds, then the loss of a configuration $c$ equals the number of gold arcs that are *not* individually reachable from $c$, which is easy to compute in most parsers.

Gómez-Rodríguez and Fernández-González (2015) show that the non-projective Covington parser is not arc-decomposable because sets of individually reachable arcs may form cycles together with already-built arcs, preventing them from being jointly reachable due to the acyclicity constraint. In spite of this, they prove that a dynamic oracle for the Covington parser can be efficiently built by counting individually unreachable arcs, and correcting for the presence of such cycles. Concretely, the loss is computed as:

$$\ell(c) = |\mathcal{U}(c, t_G)| + n_c(A \cup \mathcal{I}(c, t_G))$$

where $\mathcal{I}(c, t_G) = \{x \rightarrow y \in t_G \mid c \rightsquigarrow (x \rightarrow y)\}$ is the set of **individually reachable arcs** of $t_G$ from configuration $c$; $\mathcal{U}(c, t_G)$ is the set of **individually unreachable arcs** of $t_G$ from $c$, computed as $t_G \setminus \mathcal{I}(c, t_G)$; and $n_c(G)$ denotes the number of cycles in a graph $G$.

Therefore, to calculate the loss of a configuration $c$, we only need to compute the two terms $|\mathcal{U}(c, t_G)|$ and $n_c(A \cup \mathcal{I}(c, t_G))$. To calculate the first term, given a configuration $c$ with focus words $i$ and $j$ (i.e., $c = \langle \lambda_1|i, \lambda_2, j|B, A \rangle$), an arc $x \rightarrow y$ will be in $\mathcal{U}(c, t_G)$ if it is not in $A$, and at least one of the following holds:

- $j > \max(x, y)$, (i.e., we have read too far in the string and can no longer get $\max(x, y)$ as right focus word),
- $j = \max(x, y) \land i < \min(x, y)$, (i.e., we have $\max(x, y)$ as the right focus word but the left focus word has already moved left past $\min(x, y)$, and we cannot go back),
- there is some $z \neq 0, z \neq x$ such that $z \rightarrow y \in A$, (i.e., we cannot create $x \rightarrow y$ because it would violate the single-head constraint),
- $x$ and $y$ are on the same weakly connected component of $A$ (i.e., we cannot create $x \rightarrow y$ due to the acyclicity constraint).

The second term of the loss, $n_c(A \cup \mathcal{I}(c, t_G))$, can be computed by first obtaining $\mathcal{I}(c, t_G)$ as $t_G \setminus \mathcal{U}(c, t_G)$. Since the graph $\mathcal{I}(c, t_G)$ has in-degree 1, the algorithm by Tarjan (1972) can then be used to find and count the cycles in $O(n)$ time.

---

**Algorithm 1** Computation of the loss of a configuration in the monotonic oracle.

```
1: function LOSS(c = ⟨λ₁|i, λ₂, j|B, A⟩, t_G)
2:     U ← ∅                    ▷ Variable U is for 𝒰(c, t_G)
3:     for each x → y ∈ (t_G \ A) do
4:         left ← min(x, y)
5:         right ← max(x, y)
6:         if j > right ∨
7:             (j = right ∧ i < left) ∨
8:             (∃z > 0, z ≠ x : z → y ∈ A) ∨
9:             WEAKLYCONNECTED(A, x, y) then
10:                U ← u ∪ {x → y}
11:    I ← t_G \ U              ▷ Variable I is for ℐ(c, t_G)
12:    return |U| + COUNTCYCLES(A ∪ I)
```

---

Algorithm 1 shows the resulting loss calculation algorithm, where COUNTCYCLES is a function that counts the number of cycles in the given graph and WEAKLYCONNECTED returns whether two given nodes are weakly connected in $A$.

## 3 Non-Monotonic Transition System for the Covington Non-Projective Parser

We now define a non-monotonic variant of the Covington non-projective parser. To do so, we allow the Right-Arc and Left-Arc transitions to create arcs between any pair of nodes without restriction. If the node attached as dependent already had a previous head, the existing attachment is discarded in favor of the new one. This allows the parser to correct erroneous attachments made in the past by assigning new heads, while still enforcing the single-head constraint, as only the most recent head assigned to each node is kept.

To enforce acyclicity, one possibility would be to keep the logic of the monotonic algorithm, forbidding the creation of arcs that would create cycles. However, this greatly complicates the definition of the set of individually unreachable arcs, which is needed to compute the loss bounds that will be used by the dynamic oracle. This is because a gold arc $x \rightarrow y$ may superficially seem unreachable due to forming a cycle together with arcs in $A$, but it might in fact be reachable if there is some transition sequence that first breaks the cycle using non-monotonic transitions to remove arcs from $A$, to then create $x \rightarrow y$. We do not know of a way to characterize the conditions under which such a transition sequence exists, and thus cannot estimate the loss efficiently.

Instead, we enforce the acyclicity constraint in a similar way to the single-head constraint: Right-Arc and Left-Arc transitions are always allowed, even if the prospective arc would create a cycle in $A$. However, if the creation of a new arc $x \rightarrow y$ generates a cycle in $A$, we immediately remove the arc of the form $z \rightarrow x$ from $A$ (which trivially exists, and is unique due to the single-head constraint). This not only enforces the acyclicity constraint while keeping the computation of $\mathcal{U}(c, t_G)$ simple and efficient, but also produces a straightforward, coherent algorithm (arc transitions are always allowed, and both constraints are enforced by deleting a previous arc) and allows us to exploit non-monotonicity to the maximum (we can not only recover from assigning a node the wrong head, but also from situations where previous errors together with the acyclicity constraint prevent us from building a gold arc, keeping with the principle that later decisions override earlier ones).

In Figure 2, we can see the resulting non-monotonic transition system for the non-projective Covington algorithm, where, unlike the monotonic version, all transitions are allowed at each configuration, and the single-head and acyclicity constraints are kept in $A$ by removing offending arcs.

## 4 Non-Monotonic Approximate Dynamic Oracle

To successfully train a non-monotonic system, we need a dynamic oracle with error exploration, so that the parser will be put in erroneous states and need to apply non-monotonic transitions in order to repair them. To achieve that, we modify the dynamic oracle defined by Gómez-Rodríguez and Fernández-González (2015) so that it can deal with non-monotonicity. Our modification is an **approximate** dynamic oracle: due to the extra flexibility added to the algorithm by non-monotonicity, we do not know of an efficient way of obtaining an exact calculation of the loss of a given configuration. Instead, we use upper or lower bounds on the loss, which we empirically show to be very tight (less that 1% relative error with respect to the real loss) and are sufficient for the algorithm to provide better accuracy than the exact monotonic oracle.

First of all, we adapt the computation of the set of **individually unreachable arcs** $\mathcal{U}(c, t_G)$ to the new algorithm. In particular, if $c$ has focus words $i$ and $j$ (i.e., $c = \langle \lambda_1|i, \lambda_2, j|B, A\rangle$), then an arc $x \rightarrow y$ is in $\mathcal{U}(c, t_G)$ if it is not in $A$, and at least one of the following holds:

- $j > \max(x, y)$, (i.e., we have read too far in the string and can no longer get $\max(x, y)$ as

| | | |
|---|---|---|
| Shift: | $\langle \lambda_1, \lambda_2, j | B, A \rangle \Rightarrow \langle \lambda_1 \cdot \lambda_2 | j, [], B, A \rangle$ | |
| No-Arc: | $\langle \lambda_1 | i, \lambda_2, B, A \rangle \Rightarrow \langle \lambda_1, i | \lambda_2, B, A \rangle$ | |
| Left-Arc: | $\langle \lambda_1 | i, \lambda_2, j | B, A \rangle \Rightarrow \langle \lambda_1, i | \lambda_2, j | B, (A \cup \{j \rightarrow i\})$ | |
| | $\backslash \{x \rightarrow i \in A\} \backslash \{k \rightarrow j \in A \mid i \rightarrow^* k \in A\}\rangle$ | |
| Right-Arc: | $\langle \lambda_1 | i, \lambda_2, j | B, A \rangle \Rightarrow \langle \lambda_1, i | \lambda_2, j | B, A \cup \{i \rightarrow j\}$ | |
| | $\backslash \{x \rightarrow j \in A\} \backslash \{k \rightarrow i \in A \mid j \rightarrow^* k \in A\}\rangle$ | |

Figure 2: Transitions of the non-monotonic Covington non-projective dependency parser. The notation $i \rightarrow^* j \in A$ means that there is a (possibly empty) directed path from $i$ to $j$ in $A$.

right focus word),

- $j = \max(x, y) \wedge i < \min(x, y)$ (i.e., we have $\max(x, y)$ as the right focus word but the left focus word has already moved left past $\min(x, y)$, and we cannot move it back).

Note that, since the head of a node can change during the parsing process and arcs that produce cycles in $A$ can be built, the two last conditions present in the monotonic scenario for computing $\mathcal{U}(c, t_G)$ are not needed when we use non-monotonicity and, as a consequence, the set of **individually reachable arcs** $\mathcal{I}(c, t_G)$ is larger: due to the greater flexibility provided by non-monotonicity, we can reach arcs that would be unreachable for the monotonic version.

Since arcs that are in this new $\mathcal{U}(c, t_G)$ are unreachable even by the non-monotonic parser, $|\mathcal{U}(c, t_G)|$ is trivially a **lower bound** of the loss $\ell(c)$. It is worth noting that there always exists at least one transition sequence that builds every arc in $\mathcal{I}(c, t_G)$ at some point (although not all of them necessarily appear in the final tree, due to non-monotonicity). This can be easily shown based on the fact that the non-monotonic parser does not forbid transitions at any configuration. Thanks to this, we can can generate one such sequence by just applying the original Covington (2001) criteria (choose an arc transition whenever the focus words are linked in $\mathcal{I}(c, t_G)$, and otherwise Shift or No-Arc depending on whether the left focus word is the first word in the sentence or not), although this sequence is not necessarily optimal in terms of loss. In such a transition sequence, the gold arcs that are missed are (1) those in $\mathcal{U}(c, t_G)$, and (2) those that are removed by the cycle-breaking in Left-Arc and Right-Arc transitions. In practice configurations where (2) is needed are uncommon, so this lower bound is a very close approximation of the real loss, as will be seen empirically below.

This reasoning also helps us calculate an up-per bound of the loss: in a transition sequence as described, if we *only* build the arcs in $\mathcal{I}(c, t_G)$ and none else, the amount of arcs removed by breaking cycles (2) cannot be larger than the number of cycles in $A \cup \mathcal{I}(c, t_G)$. This means that $|\mathcal{U}(c, t_G)| + n_c(A \cup \mathcal{I}(c, t_G))$ is an **upper bound** of the loss $\ell(c)$. Note that, contrary to the monotonic case, this expression does not always give us the exact loss, for several reasons: firstly, $A \cup \mathcal{I}(c, t_G)$ can have non-disjoint cycles (a node may have different heads in $A$ and $\mathcal{I}$ since attachments are not permanent, contrary to the monotonic version) and thus removing a single arc may break more than one cycle; secondly, the removed arc can be a non-gold arc of $A$ and therefore not incur loss; and thirdly, there may exist alternative transition sequences where a cycle in $A \cup \mathcal{I}(c, t_G)$ is broken early by non-monotonic configurations that change the head of a wrongly-attached node in $A$ to a different (and also wrong) head,[3] removing the cycle before the cycle-breaking mechanism needs to come into play without incurring in extra errors. Characterizing the situations where such an alternative exists is the main difficulty for an exact calculation of the loss.

However, it is possible to obtain a closer upper bound to the real loss if we consider the following: for each cycle in $A \cup \mathcal{I}(c, t_G)$ that will be broken by the transition sequence described above, we can determine exactly which is the arc removed by cycle-breaking (if $x \rightarrow y$ is the arc that will close the cycle according to the Covington arc-building order, then the affected arc is the one of the form $z \rightarrow x$). The cycle can only cause the loss of a gold arc if that arc $z \rightarrow x$ is gold, which can be trivially checked. Hence, if we call cycles where that holds **problematic cycles**, then the expression

---

[3] Note that, in this scenario, the new head must also be wrong because otherwise the newly created arc would be an arc of $\mathcal{I}(c, t_G)$ (and therefore, would not be breaking a cycle in $A \cup \mathcal{I}(c, t_G)$). However, replacing a wrong attachment with another wrong attachment need not increase loss.

| Language | average value | | | | relative difference to loss | | |
|---|---|---|---|---|---|---|---|
| | lower | loss | pc upper | upper | lower | pc upper | upper |
| Arabic | 0.66925 | 0.67257 | **0.67312** | 0.68143 | 0.00182 | **0.00029** | 0.00587 |
| Basque | **0.58260** | 0.58318 | 0.58389 | 0.62543 | **0.00035** | 0.00038 | 0.02732 |
| Catalan | 0.58009 | 0.58793 | **0.58931** | 0.60644 | 0.00424 | **0.00069** | 0.00961 |
| Chinese | **0.56515** | 0.56711 | 0.57156 | 0.62921 | **0.00121** | 0.00302 | 0.03984 |
| Czech | **0.57521** | 0.58357 | 0.59401 | 0.62883 | **0.00476** | 0.00685 | 0.02662 |
| English | 0.55267 | 0.56383 | **0.56884** | 0.59494 | 0.00633 | **0.00294** | 0.01767 |
| Greek | 0.56123 | 0.57443 | **0.57983** | 0.61256 | 0.00731 | **0.00296** | 0.02256 |
| Hungarian | **0.46495** | 0.46672 | 0.46873 | 0.48797 | **0.00097** | 0.00114 | 0.01165 |
| Italian | 0.62033 | 0.62612 | **0.62767** | 0.64356 | 0.00307 | **0.00082** | 0.00883 |
| Turkish | **0.60143** | 0.60215 | 0.60660 | 0.63560 | **0.00060** | 0.00329 | 0.02139 |
| Bulgarian | 0.61415 | 0.62257 | **0.62433** | 0.64497 | 0.00456 | **0.00086** | 0.01233 |
| Danish | 0.67350 | 0.67904 | **0.68119** | 0.69436 | 0.00291 | **0.00108** | 0.00916 |
| Dutch | 0.69201 | 0.70600 | **0.71105** | 0.74008 | 0.00709 | **0.00251** | 0.01862 |
| German | **0.54581** | 0.54755 | 0.55080 | 0.58182 | **0.00104** | 0.00208 | 0.02033 |
| Japanese | **0.60515** | 0.60515 | **0.60515** | 0.60654 | **0.00000** | **0.00000** | 0.00115 |
| Portuguese | 0.58880 | 0.60063 | **0.60185** | 0.61780 | 0.00651 | **0.00067** | 0.00867 |
| Slovene | 0.56155 | 0.56860 | **0.57135** | 0.60373 | 0.00396 | **0.00153** | 0.01979 |
| Spanish | 0.58247 | 0.59119 | **0.59277** | 0.61273 | 0.00487 | **0.00089** | 0.01197 |
| Swedish | 0.57543 | 0.58636 | **0.58933** | 0.61104 | 0.00585 | **0.00153** | 0.01383 |
| Average | 0.59009 | 0.59656 | **0.59954** | 0.62416 | 0.00355 | **0.00176** | 0.01513 |

Table 1: Average value of the different bounds and the loss, and of the relative differences from each bound to the loss, on CoNLL-XI (first block) and CoNLL-X (second block) datasets during 100,000 transitions. For each language, we show in boldface the average value and relative difference of the bound that is closer to the loss.

$|\mathcal{U}(c, t_G)| + n_{pc}(A \cup \mathcal{I}(c, t_G))$, where "pc" stands for problematic cycles, is a closer **upper bound** to the loss $\ell(c)$ and the following holds:

$$|\mathcal{U}(c, t_G)| \le \ell(c) \le |\mathcal{U}(c, t_G)| + n_{pc}(A \cup \mathcal{I}(c, t_G))$$

$$\le |\mathcal{U}(c, t_G)| + n_c(A \cup \mathcal{I}(c, t_G))$$

As mentioned before, unlike the monotonic approach, a node can have a different head in $A$ than in $\mathcal{I}(c, t_G)$ and, as a consequence, the resulting graph $A \cup \mathcal{I}(c, t_G)$ has maximum in-degree 2 rather than 1, and there can be overlapping cycles. Therefore, the computation of the non-monotonic terms $n_c(A \cup \mathcal{I}(c, t_G))$ and $n_{pc}(A \cup \mathcal{I}(c, t_G))$ requires an algorithm such as the one by Johnson (1975) to find all elementary cycles in a directed graph. This runs in $O((n + e)(c + 1))$, where $n$ is the number of vertices, $e$ is the number of edges and $c$ is the number of elementary cycles in the graph. This implies that the calculation of the two non-monotonic upper bounds is less efficient than the linear loss computation in the monotonic scenario. However, a non-monotonic algorithm that uses the lower bound as loss expression is the fastest option (even faster than the monotonic approach) as the oracle does not need to compute cycles at all, speeding up the training process.

Algorithm 2 shows the non-monotonic variant of Algorithm 1, where COUNTRELEVANT-CYCLES is a function that counts the number of cycles or problematic cycles in the given graph,

---

**Algorithm 2** Computation of the approximate loss of a non-monotonic configuration.

1: **function** LOSS($c = \langle \lambda_1 | i, \lambda_2, j | B, A \rangle, t_G$)
2: $U \leftarrow \emptyset$ ▷ Variable U is for $\mathcal{U}(c, t_G)$
3: **for each** $x \rightarrow y \in (t_G \setminus A)$ **do**
4: $left \leftarrow \min(x, y)$
5: $right \leftarrow \max(x, y)$
6: **if** $j > right \vee$
7: $(j = right \wedge i < left)$ **then**
8: $U \leftarrow u \cup \{x \rightarrow y\}$
9: $I \leftarrow t_G \setminus U$ ▷ Variable I is for $\mathcal{I}(c, t_G)$
10: **return** $|U| + $ COUNTRELEVANTCYCLES$(A \cup I)$

---

depending on the upper bound implemented, and will return 0 in case we use the lower bound.

## 5 Evaluation of the loss bounds

To determine how close the lower bound $|\mathcal{U}(c, t_G)|$ and the upper bounds $|\mathcal{U}(c, t_G)| + n_{pc}(A \cup \mathcal{I}(c, t_G))$ and $|\mathcal{U}(c, t_G)| + n_c(A \cup \mathcal{I}(c, t_G))$ are to the actual loss in practical scenarios, we use exhaustive search to calculate the real loss of a given configuration, to then compare it with the bounds. This is feasible because the lower and upper bounds allow us to prune the search space: if an upper and a lower bound coincide for a configuration we already know the loss and need not keep searching, and if we can branch to two configurations such that the lower bound of one is greater or equal than an upper bound of the other, we can discard the former as it will never lead to smaller loss than the latter. Therefore, this ex-

| Unigrams |
|---|
| $L_0w$; $L_0p$; $L_0wp$; $L_0l$; $L_{0h}w$; $L_{0h}p$; $L_{0h}l$; $L_{0l'}w$; $L_{0l'}p$; $L_{0l'}l$; $L_{0r'}w$; $L_{0r'}p$; $L_{0r'}l$; $L_{0h2}w$; $L_{0h2}p$; $L_{0h2}l$; $L_{0l}w$; $L_{0l}p$; $L_{0l}l$; $L_{0r}w$; $L_{0r}p$; $L_{0r}l$; $L_0wd$; $L_0pd$; $L_0wv_r$; $L_0pv_r$; $L_0wv_l$; $L_0pv_l$; $L_0ws_l$; $L_0ps_l$; $L_0ws_r$; $L_0ps_r$; $L_1w$; $L_1p$; $L_1wp$; $R_0w$; $R_0p$; $R_0wp$; $R_{0h}w$; $R_{0h}p$;$R_{0h}l$; $R_{0h2}w$; $R_{0h2}p$; $R_{0l'}w$; $R_{0l'}p$; $R_{0l'}l$; $R_{0l}w$; $R_{0l}p$; $R_{0l}l$; $R_0wd$; $R_0pd$; $R_0wv_l$; $R_0pv_l$; $R_0ws_l$; $R_0ps_l$; $R_1w$; $R_1p$; $R_1wp$; $R_2w$; $R_2p$; $R_2wp$; $CLw$; $CLp$; $CLwp$; $CRw$; $CRp$; $CRwp$; |
| **Pairs** |
| $L_0wp+R_0wp$; $L_0wp+R_0w$; $L_0w+R_0wp$; $L_0wp+R_0p$; $L_0p+R_0wp$; $L_0w+R_0w$; $L_0p+R_0p$; $R_0p+R_1p$; $L_0w+R_0wd$; $L_0p+R_0pd$; |
| **Triples** |
| $R_0p+R_1p+R_2p$; $L_0p+R_0p+R_1p$; $L_{0h}p+L_0p+R_0p$; $L_0p+L_{0l'}p+R_0p$; $L_0p+L_{0r'}p+R_0p$; $L_0p+R_0p+R_{0l'}p$; $L_0p+L_{0l'}p+L_{0l}p$; $L_0p+L_{0r'}p+L_{0r}p$; $L_0p+L_{0h}p+L_{0h2}p$; $R_0p+R_{0l'}p+R_{0l}p$; |

Table 2: Feature templates. $L_0$ and $R_0$ denote the left and right focus words; $L_1, L_2, \ldots$ are the words to the left of $L_0$ and $R_1, R_2, \ldots$ those to the right of $R_0$. $X_{ih}$ means the head of $X_i$, $X_{ih2}$ the grandparent, $X_{il}$ and $X_{il'}$ the farthest and closest left dependents, and $X_{ir}$ and $X_{ir'}$ the farthest and closest right dependents, respectively. $CL$ and $CR$ are the first and last words between $L_0$ and $R_0$ whose head is not in the interval $[L_0, R_0]$. Finally, $w$ stands for word form; $p$ for PoS tag; $l$ for dependency label; $d$ is the distance between $L_0$ and $R_0$; $v_l$, $v_r$ are the left/right valencies (number of left/right dependents); and $s_l$, $s_r$ the left/right label sets (dependency labels of left/right dependents).

haustive search with pruning guarantees to find the exact loss.

Due to the time complexity of this process, we undertake the analysis of only the first 100,000 transitions on each dataset of the nineteen languages available from CoNLL-X and CoNLL-XI shared tasks (Buchholz and Marsi, 2006; Nivre et al., 2007). In Table 1, we present the average values for the lower bound, both upper bounds and the loss, as well as the relative differences from each bound to the real loss. After those experiments, we conclude that the lower and the closer upper bounds are a tight approximation of the loss, with both bounds incurring relative errors below 0.8% in all datasets. If we compare them, the real loss is closer to the upper bound $|\mathcal{U}(c,t_G)| + n_{pc}(A \cup \mathcal{I}(c,t_G))$ in the majority of datasets (12 out of 18 languages, excluding Japanese where both bounds were exactly equal to the real loss in the whole sample of configurations). This means that the term $n_{pc}(A \cup \mathcal{I}(c,t_G))$ provides a close approximation of the gold arcs missed by the presence of cycles in $A$. Regarding the upper bound $|\mathcal{U}(c,t_G)| + n_c(A \cup \mathcal{I}(c,t_G))$,

it presents a more variable relative error, ranging from 0.1% to 4.0%.

Thus, although we do not know an algorithm to obtain the exact loss which is fast enough to be practical, any of the three studied loss bounds can be used to obtain a feasible approximate dynamic oracle with full non-monotonicity.

## 6 Experiments

To prove the usefulness of our approach, we implement the static, dynamic monotonic and non-monotonic oracles for the non-projective Covington algorithm and compare their accuracies on nine datasets from the CoNLL-X shared task (Buchholz and Marsi, 2006) and all datasets from the CoNLL-XI shared task (Nivre et al., 2007). For the non-monotonic algorithm, we test the three different loss expressions defined in the previous section. We train an averaged perceptron model for 15 iterations and use the same feature templates for all languages, which are listed in detail in Table 2.

The accuracies obtained by the non-projective Covington parser with the three available oracles are presented in Table 3. For the non-monotonic dynamic oracle, three variants are shown, one for each loss expression implemented. As we can see, the novel non-monotonic oracle improves over the accuracy of the monotonic version on 14 out of 19 languages (0.32 in UAS on average) with the best loss calculation $|\mathcal{U}(c,t_G)| + n_c(A \cup \mathcal{I}(c,t_G))$, where 6 of these improvements are statistically significant at the .05 level (Yeh, 2000). The other two loss calculation methods also achieve good results, outperforming the monotonic algorithm on 12 out of 19 datasets tested.

The loss expression $|\mathcal{U}(c,t_G)| + n_c(A \cup \mathcal{I}(c,t_G))$ obtains greater accuracy on average than the other two loss expressions, including the more adjusted upper bound that is provably closer to the real loss. This could be explained by the fact that identifying problematic cycles is a difficult task to learn for the parser, and for this reason a more straightforward approach, which tries to avoid all kinds of cycles (regardless of whether they will cost gold arcs or not), can perform better. This also leads us to hypothesize that, even if it were feasible to build an oracle with the exact loss, it would not provide practical improvements over these approximate oracles; as it appears difficult for a statistical model to learn the situations where repla-

| | static | | dynamic monotonic | | dynamic non-monotonic lower | | pc upper | | upper | |
|---|---|---|---|---|---|---|---|---|---|---|
| Language | UAS | LAS | UAS | LAS | UAS | LAS | UAS | LAS | UAS | LAS |
| Arabic | 80.67 | 66.51 | 82.76* | 68.48* | 83.29* | 69.14* | 83.18* | 69.05* | 83.40$^\dagger$ | 69.29$^\dagger$ |
| Basque | 76.55 | 66.05 | **77.49**$^\dagger$ | **67.31**$^\dagger$ | 74.61 | 65.31 | 74.69 | 65.18 | 74.27 | 64.78 |
| Catalan | 90.52 | 85.09 | **91.37*** | **85.98*** | 90.51 | 85.35 | 90.40 | 85.30 | 90.44 | 85.35 |
| Chinese | 84.93 | 80.80 | 85.82 | 82.15 | 86.55* | **82.53*** | 86.29* | 82.27* | **86.60*** | 82.51* |
| Czech | 78.49 | 61.77 | 80.21* | 63.52* | 81.32$^\dagger$ | 64.89$^\dagger$ | 81.33$^\dagger$ | 64.81$^\dagger$ | 81.49$^\dagger$ | 65.18$^\dagger$ |
| English | 85.35 | 84.29 | 87.47* | 86.55* | 88.44$^\dagger$ | 87.37$^\dagger$ | 88.23$^\dagger$ | 87.22$^\dagger$ | 88.50$^\dagger$ | 87.55$^\dagger$ |
| Greek | 79.47 | 69.35 | 80.76 | 70.43 | 80.90 | 70.46 | 80.84 | 70.34 | 81.02* | 70.49* |
| Hungarian | 77.65 | 68.32 | **78.84*** | **70.16*** | 78.67* | 69.83* | 78.47* | 69.66* | 78.65* | 69.74* |
| Italian | 84.06 | 79.79 | 84.30 | 80.17 | 84.38 | 80.30 | **84.64** | **80.52** | 84.47 | 80.32 |
| Turkish | **81.28** | 70.97 | 81.14 | **71.38** | 80.65 | 71.15 | 80.80 | 71.29 | 80.60 | 71.07 |
| Bulgarian | 89.13 | 85.30 | 90.45* | 86.86* | 91.36$^\dagger$ | 87.88$^\dagger$ | 91.33$^\dagger$ | 87.89$^\dagger$ | **91.73**$^\dagger$ | **88.26**$^\dagger$ |
| Danish | 86.00 | 81.49 | 86.91* | **82.75*** | 86.83* | 82.63* | 86.89* | 82.74* | **86.94*** | 82.68* |
| Dutch | 81.54 | 78.46 | 82.07 | 79.26 | 82.78* | 79.64* | 82.80* | 79.68* | **83.02**$^\dagger$ | **79.92**$^\dagger$ |
| German | 86.97 | 83.91 | **87.95*** | **85.17*** | 87.31 | 84.37 | 87.18 | 84.22 | 87.48 | 84.54 |
| Japanese | 93.63 | 92.20 | 93.67 | 92.33 | **94.02** | **92.68** | **94.02** | **92.68** | 93.97 | 92.66 |
| Portuguese | 86.55 | 82.61 | **87.45*** | 83.62* | 87.17* | 83.47* | 87.12* | 83.45* | 87.40* | **83.71*** |
| Slovene | 76.76 | 63.53 | 77.86 | 64.43 | 80.39$^\dagger$ | 67.04$^\dagger$ | **80.56**$^\dagger$ | 67.10$^\dagger$ | 80.47$^\dagger$ | 67.10$^\dagger$ |
| Spanish | 79.20 | 76.00 | 80.12* | 77.24* | **81.36**$^\dagger$ | **78.30**$^\dagger$ | 81.12* | 77.99* | 81.33* | 78.16* |
| Swedish | 87.43 | 81.77 | 88.05* | 82.77* | 88.20* | 83.02* | 88.09* | 82.87* | **88.36*** | **83.16*** |
| Average | 83.48 | 76.75 | 84.46 | 77.92 | 84.67 | 78.18 | 84.63 | 78.12 | **84.74** | **78.24** |

Table 3: Parsing accuracy (UAS and LAS, including punctuation) of the Covington non-projective parser with static, and dynamic monotonic and non-monotonic oracles on CoNLL-XI (first block) and CoNLL-X (second block) datasets. For the dynamic non-monotonic oracle, we show the performance with the three loss expressions, where *lower* stands for the lower bound $|\mathcal{U}(c, t_G)|$, *pc upper* for the upper bound $|\mathcal{U}(c, t_G)| + n_{pc}(A \cup \mathcal{I}(c, t_G))$, and *upper* for the upper bound $|\mathcal{U}(c, t_G)| + n_c(A \cup \mathcal{I}(c, t_G))$. For each language, we run five experiments with the same setup but different seeds and report the averaged accuracy. Best results for each language are shown in boldface. Statistically significant improvements ($\alpha = .05$) of both dynamic oracles are marked with * if they are only over the static oracle, and with $^\dagger$ if they are over the opposite dynamic oracle too.

cing a wrong arc with another indirectly helps due to breaking prospective cycles.

It is also worth mentioning that the non-monotonic dynamic oracle with the best loss expression accomplishes an average improvement over the static version (1.26 UAS) greater than that obtained by the monotonic oracle (0.98 UAS), resulting in 13 statistically significant improvements achieved by the non-monotonic variant over the static oracle in comparison to the 12 obtained by the monotonic system. Finally, note that, despite this remarkable performance, the non-monotonic version (regardless of the loss expression implemented) has an inexplicable drop in accuracy in Basque in comparison to the other two oracles.

## 7 Conclusion

We presented a novel, fully non-monotonic variant of the well-known non-projective Covington parser, trained with a dynamic oracle. Due to the unpredictability of a non-monotonic scenario, the real loss of each configuration cannot be computed. To overcome this, we proposed three different loss expressions that closely bound the loss and enable us to implement a practical non-monotonic dynamic oracle.

On average, our non-monotonic algorithm obtains better performance than the monotonic version, regardless of the loss calculation used. In particular, one of the loss expressions developed proved very promising by providing the best average accuracy, in spite of being the farthest approximation from the actual loss. On the other hand, the proposed lower bound makes the non-monotonic system the fastest one among all dynamic oracles developed for the non-projective Covington algorithm.

To our knowledge, this is the first implementation of non-monotonicity for a non-projective parsing algorithm, and the first approximate dynamic oracle that uses close, efficiently-computable approximations of the loss, showing this to be a feasible alternative when it is not practical to compute the actual loss.

While we used a perceptron classifier for our experiments, our oracle could also be used in neural-network implementations of greedy transition-based parsing (Chen and Manning, 2014; Dyer et al., 2015), providing an interesting avenue for future work.

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
