# Peer review of "A Full Non-Monotonic Transition System for Unrestricted Non-Projective Parsing"

_ACL 2017 — decision unknown_

[Official Review · Reviewer 1 · rating 4 · confidence 5]
soundness 4 · originality 3 · clarity 5 · impact 3 · substance 4 · appropriateness 5 · meaningful comparison 4 · presentation format Oral Presentation

- Strengths:

The paper makes several novel contributions to (transition-based) dependency
parsing by extending the notion of non-monotonic transition systems and dynamic
oracles to unrestricted non-projective dependency parsing. The theoretical and
algorithmic analysis is clear and insightful, and the paper is admirably clear.

- Weaknesses:

Given that the main motivation for using Covington's algorithm is to be able to
recover non-projective arcs, an empirical error analysis focusing on
non-projective structures would have further strengthened the paper. And even
though the main contributions of the paper are on the theoretical side, it
would have been relevant to include a comparison to the state of the art on the
CoNLL data sets and not only to the monotonic baseline version of the same
parser.

- General Discussion:

The paper extends the transition-based formulation of Covington's dependency
parsing algorithm (for unrestricted non-projective structures) by allowing
non-monotonicity in the sense that later transitions can change structure built
by earlier transitions. In addition, it shows how approximate dynamic oracles
can be formulated for the new system. Finally, it shows experimentally that the
oracles provide a tight approximation and that the non-monotonic system leads
to improved parsing accuracy over its monotonic counterpart for the majority of
the languages included in the study.

The theoretical contributions are in my view significant enough to merit
publication, but I also think the paper could be strengthened on the empirical
side. In particular, it would be relevant to investigate, in an error analysis,
whether the non-monotonic system improves accuracy specifically on
non-projective structures. Such an analysis can be motivated on two grounds:
(i) the ability to recover non-projective structures is the main motivation for
using Covington's algorithm in the first place; (ii) non-projective structures
often involved long-distance dependencies that are hard to predict for a greedy
transition-based parser, so it is plausible that the new system would improve
the situation. 

Another point worth discussion is how the empirical results relate to the state
of the art in light of recent improvements thanks to word embeddings and neural
network techniques. For example, the non-monotonicity is claimed to mitigate
the error propagation typical of classical greedy transition-based parsers. But
another way of mitigating this problem is to use recurrent neural networks as
preprocessors to the parser in order to capture more of the global sentence
context in word representations. Are these two techniques competing or
complementary? A full investigation of these issues is clearly outside the
scope of the paper, but some discussion would be highly relevant.

Specific questions:

Why were only 9 out of the 13 data sets from the CoNLL-X shared task used? I am
sure there is a legitimate reason and stating it explicitly may prevent readers
from becoming suspicious. 

Do you have any hypothesis about why accuracy decreases for Basque with the
non-monotonic system? Similar (but weaker) trends can be seen also for Turkish,
Catalan, Hungarian and (perhaps) German.

How do your results compare to the state of the art on these data sets? This is
relevant for contextualising your results and allowing readers to estimate the
significance of your improvements.

Author response:

I am satisfied with the author's response and see no reason to change my
previous review.